

# Validating MODIS Above-cloud Aerosol Optical Depth Retrieved from 'Color Ratio' Algorithm using Direct Measurements made by NASA's Airborne AATS and 4STAR Sensors

Hiren Jethva[1,2], Omar Torres[2], Lorraine Remer[3], Jens Redemann[4], John Livingston[5], Stephen Dunagan[4], Yohei Shinozuka[6], Meloe Kacenelenbogen[6], Michal Segal Rosenheimer[6], and Rob Spurr[7]

[1]Universities Space Research Association, Columbia, MD 21044 USA
[2]NASA Goddard Space Flight Center, Greenbelt, MD 20771 USA
[3]University of Maryland, Baltimore County, Baltimore, MD 21250 USA
[4]NASA Ames Research Center, Moffett Field, CA 94035, USA
[5]SRI International, Menlo Park, CA 94025, USA
[6]NASA ARC-CREST, Moffett Field, CA 94035, USA
[7]RT Solutions, Cambridge, MA 02138, USA

*Correspondence to:* Hiren Jethva (hiren.t.jethva@nasa.gov)

**Abstract.** We present the first ever validation of above-cloud aerosol optical depth (ACAOD) retrieved from the 'color ratio' method applied to MODIS cloudy-sky reflectance measurements using the limited direct measurements made by NASA's airborne AATS and 4STAR sensors. A thorough search of the airborne database collection revealed a total of five events in which airborne sunphotometer, coincident with the MODIS overpass, observed agricultural biomass burning, dust, and

5   wildfire-emitted aerosols above a low-level cloud deck during SAFARI-2000, ACE-ASIA 2001, and SEAC4RS-2013 campaigns, respectively. The co-located satellite-airborne measurements revealed a good agreement (root-mean-square-error<0.1) with most matchups falling within the estimated uncertainties associated the MODIS retrievals (about -10% to +50%). The co-retrieved cloud optical depth was comparable to that of the MODIS operational cloud product for ACE-ASIA and SEAC4RS, however, higher by 30%-50% for the SAFARI-2000 case study. The reason for this discrepancy could be attributed to the

10   distinct aerosol optical properties encountered during respective campaigns. A brief discussion on the overall uncertainty in the satellite-based ACAOD retrieval is presented. Field experiments dedicated to the direct measurements of aerosols above cloud are needed for the extensive validation of satellite-based retrievals.

Keywords: Above-cloud Aerosol Optical Depth, Validation, MODIS, AATS, 4STAR



# 1 Introduction

Aerosol-cloud interaction continues to be one of the leading uncertain components of climate models, primarily due to the lack of an adequate knowledge of the complex microphysical and radiative processes associated with aerosol-cloud system (Stocker et al., 2013). One important aspect of the problem is when aerosol and clouds are found in the same atmospheric

column, for instance, when light-absorbing aerosols such as biomass burning generated carbonaceous particles or wind-blown dust overlay low-level cloud decks. Contrary to the cloud-free scenario over dark surface, for which these aerosols are known to produce a net cooling effect (negative radiative forcing) on climate, the overlapping situation of absorbing aerosols over cloud can potentially exert a significant level of atmospheric absorption and produces a positive radiative forcing at top-of-atmosphere (TOA) (Keil and Haywood, 2003; Chand et al., 2009). The magnitude of direct radiative effects of aerosols above

cloud depends directly on the aerosol loading, and the microphysical-optical properties of the aerosol layer and the underlying cloud deck (Meyer et al., 2013) as well as geometric cloud fraction (Chand et al., 2009). The resultant aerosol-driven atmospheric heating can have a great influence on the atmospheric stability, cloud formation and lifetime, and the hydrological cycle.

In the past few years, the development of several independent algorithms that quantify aerosol loading above cloud from

satellite-based active as well passive measurements has been a major breakthrough in the aerosol and cloud remote sensing communities. These algorithms have shown the potential to retrieve quantitative information on aerosol loading above cloud using measurements from different A-train sensors including CALIPSO/CALIOP (Hu et al., 2007; Chand et al., 2008), Parasol/POLDER (Waquet et al., 2009, 2013; Peers et al., 2015), Aura/OMI (Torres et al., 2012), and Aqua/MODIS (Jethva et al., 2013; Meyer et al., 2015). Of particularl interest in the present paper is the 'color ratio' (CR) technique that retrieves

above-cloud aerosol optical depth (ACAOD) and aerosol-corrected cloud optical depth (COD), simultaneously, using OMI and MODIS observations independently. The technique is based physically on the unambiguous reduction of the ultraviolet (UV), visible (VIS), and shortwave-infrared (SWIR) radiation reaching the top-of-atmosphere, due to enhanced particle absorption above cloud. The effects of aerosol absorption have a spectral signature, in which the absorption strength is found to be stronger at shorter wavelengths than at longer. This produces a strong color effect in two-channel measurements, hence, the name 'color

ratio' method. This technique was originally developed and successfully demonstrated for OMI's near-UV measurements (354 and 388 nm) (Torres et al., 2012); it was subsequently extended to the visible-near-infrared measurements (470-860 nm) made by MODIS (Jethva et al., 2013).

Although there is unprecedented quantitative information on aerosol loading above cloud now available from A-train sen-

sors, an important question remains: How do we validate the satellite retrievals of ACAOD? Unlike the validation of cloud-free aerosol retrievals from satellites, for which ample ground-based measurements are available, validation of ACAOD is a challenging task primarily due to the lack of adequate direct measurements of aerosols in cloudy skies, specifically of aerosols above cloud. The availability of research-level retrievals of ACAOD from multiple sensors on the A-train satellite constellation offers an opportunity to inter-compare aerosol loading derived using independent techniques applied to different sensors. Jethva



et al. (2014) carried out an inter-sensor comparison analysis using retrievals from MODIS, CALIOP, POLDER, and OMI for the two scenarios of smoke above cloud observed over the southeastern Atlantic Ocean; they found that A-train sensors agree with each other to within an AOD difference of less than 0.2 over homogeneous opaque cloud fields. Despite the fact that each method is designed independently and relies on different types of measurements from different sensors, the overall close agreement between them was an encouraging result. However, the inter-sensor comparison analysis does not constitute a validation exercise; instead the level of agreement can be interpreted as a measure of consistency (or lack thereof) in the retrieval.

While ground-based measurements such as those from AERONET-Aerosol Robotic Network cannot be helpful in our situation, airborne measurements taken when the aircraft is flying above cloud seem to be the only source to validate the above-cloud aerosol retrievals. In pursuit of finding the right dataset, we have looked at the data archive of past field campaigns with a focus on aircraft-based direct measurements of AOD. We found that the airborne measurements made by NASA's 6-14-channel Ames Airborne Tracking Sunphotometer (AATS-6, -14) and their next generation sensor Spectrometer for Sky-Scanning, Sun-Tracking Atmospheric Research (4STAR) provide a valuable database for validating the satellite retrieval of ACAOD. The High Spectral Resolution Lidar (HSRL) is another instrument which can measure the vertical profile of particulate extinction without assuming a lidar ratio, and thus can provide a direct measure of AOD above cloud provided that the HSRL flies above the aerosol layer. Kacenelenbogen et al. (2014) have used HSRL measurements collected during different flights over North America for evaluating the CALIOP standard product of AOD above cloud.

In this paper, we present the validation analysis of ACAOD retrieved from the MODIS sensor using cloudy-sky airborne measurements of AOD made by AATS and 4STAR. To our knowledge, this is the first attempt to assess the accuracy of satellite-based retrieval of aerosols above cloud. We mention here that we were unable to perform a validation analysis of OMI ACAOD retrieval due to two reasons: 1) four out of the five events of aerosols above clouds observed by AATS occurred prior to the launch of OMI in 2004, and 2) an event of wildfire smoke aerosols above clouds measured by 4STAR on Aug 06, 2013 was missed by OMI due to row anomaly contamination encountered with post-2008 operation. The paper is organized as follows: Section 2 introduces datasets and the co-location approach; results of satellite versus airborne measurements are presented in section 3; and a brief discussion of the uncertainties and future scope of validation study is presented in section 4.

## 2   Datasets

### 2.1   AMES AIRBORNE TRACKING SUNPHOTOMETER (AATS)

The 6- and 14-channel AATS developed by the NASA Ames research group (Russell et al., 1993; Redemann et al., 2003) measures the transmission of the solar beam in distinct spectral bands from near-UV to visible to near-infrared and subsequently calculates the columnar aerosol optical depths (AOD). AATS's azimuth and elevation motors controlled with a quadrant differential photodiode sun sensor rotate a tracking head, which locks the detector normal to the solar beam, and thus provides the direct measurements of solar transmission. The tracking head can be mounted outside the aircraft body to minimize block-



ages by aircraft structures; this enables direct measurements of AOD during flight operation at different altitudes. Further information on AATS and its next generation sensor 4STAR can be found in the above papers as well as at the web link: http://geo.arc.nasa.gov/sgg/AATS-website/.

AATS has been operated in several field campaigns, starting as early as July 1996 during the Tropospheric Aerosol Radiative Forcing Observational Experiment (TARFOX), and including the second Aerosol Characterization Experiment (ACE-2), South African Regional Science Initiative (SAFARI) 2000 (Schmid et al., 2003b), Asian Pacific Regional Aerosol Characterization Experiment (ACE-Asia) (Schmid et al., 2003a; Redemann et al., 2003), and Chesapeake Lighthouse and Aircraft Measurements for Satellites (CLAMS) (Redemann et al., 2001). More recently, the Ames Sunphotometer/Satellite team has

developed an advanced instrument called Spectrometers for Sky-Scanning, Sun-Tracking Atmospheric Research (4STAR) (Dunagan et al., 2013; Shinozuka et al.; Segal-Rosenheimer et al., 2014), which extends the capabilities of AATS by adding a sky-scanning mechanism that enables the retrieval of complex refractive index, shape, and aerosol size distribution. Furthermore, an additional use of spectrometer can make measurements of trace gases (e.g., NO2) in order to enhance the accuracy of aerosol measurements via improved aerosol-gas separation. Recently, the 4STAR instrument participated in the Studies of

Emissions, Atmospheric Composition, Clouds and Climate Coupling by Regional Surveys (SEAC4RS) experiment conducted during August 2013 over the Southern and Western Unites States. The first flight of SEAC4RS covered parts of Oregon, California, and the neighboring Pacific Ocean and measured the properties of aerosols and trace gases emitted from wildfires over the region.

A thorough search of the AATS and 4STAR datasets has revealed a total of five significant events of aerosols above cloud observed during different field campaigns that are also co-located with the MODIS overpasses. These include an event with carbonaceous aerosols overlaying a marine boundary layer stratocumulus cloud deck over the southeastern Atlantic Ocean on Sept 13, 2000 (AATS Flight no. 1837, (Schmid et al., 2003a)), three events of dust plumes above clouds observed during ACE-ASIA in 2001 (Apr 20, 30 and May 4, (Redemann et al., 2003)) over the Sea of Japan and the East China Sea, and a

wildfire smoke event observed on Aug 06, 2013 over the Pacific Ocean during the first test flight just before the main phase of SEAC4RS-2013 began over the southern U.S.A. The absolute error in the aircraft-mounted AATS and 4STAR measurements of AOD was mostly in the range 0.02 to 0.03 with maximum error reaches up to 0.05 for certain measurements. Owing to its high accuracy relative to the expected uncertainty in the satellite retrieval, we treat aircraft measurements as truth for validating satellite retrievals of ACAOD. We use only quality-controlled, cloud-sky AOD data collected by both airborne sensors in the

present analysis.

### 2.1.1  MODIS

The presence of an absorbing aerosol layer above cloud reduces the TOA reflectance and color ratio between visible and SWIR wavelengths. The general CR technique developed by Jethva et al. (2013) exploits this unambiguous signal and uses reflectance at two channels (470 nm and 860 nm) to retrieve ACAOD and the underlying COD, simultaneously. The CR



technique was originally developed for retrieving ACAOD using OMI's near-UV observations (Torres et al., 2012); however, it was then extended to MODIS visible observations. The method requires MODIS TOA reflectances (MOD/MYD021KM), geo-location data (MOD/MYD03), and the MODIS cloud product (MOD/MYD06)–all three datasets correspond to 1-km spatial resolution. These products were obtained from http://ladsweb.nascom.nasa.gov/data/. The aerosol optical and microphysical

models required to generate look-up-tables (LUTs) of simulated TOA reflectances were derived from the multi-year statistics of the AERONET cloud-free Level 2 direct measurements and inversions carried out at Mongu (15°S, 23°E), Zambia in southern Africa, Noto (37.33°N, 137.14°E) in Japan, and HJAndrews (44.24°N, 122.22°W) in California, for representing aerosol properties for the SAFARI-2000, ACE-ASIA, and SEAC4RS campaigns, respectively. Table 1 lists the assumed aerosol microphysical and optical properties derived from AERONET dataset for the three campaigns. We employ the VLIDORT V2.6

polarized radiative transfer model [Spurr, 2006] for the simulation of LUT reflectances. Aerosol vertical profile is assumed to follow a Gaussian distribution with the peak at height 3 km; a cloud layer was placed between 1 and 1.5 km–both are generally consistent with the climatological vertical structure of aerosols and clouds observed by the CALIOP lidar over the three regions. The retrieved ACAOD at 470/860 nm was converted to its value at 500 nm according to the spectral extinction assumed in the selected aerosol models.

**2.2   Co-location of Satellite-Airborne Sensors**

In contrast to the validation of cloud-free AOD retrieved from satellite, in which columnar retrievals are compared against ground-based measurements following a static spatio-temporal approach (Ichoku et al., 2002), validating above-cloud aerosol retrieval using airborne data poses different challenges. First, the airborne sensor is on a moving platform both in horizontal and vertical directions; it therefore needs to be continuously tracked for the co-location with nearby satellite retrieval. Second,

it makes measurements at different altitudes along the aircraft trajectory, which more often than not, do not represent the same atmospheric column above the cloud that is seen from the satellite. In order to make the satellite-airborne measurements comparable, therefore, either the airborne measurements need to be adjusted all the way down to the cloud top, or else the satellite retrieval must be scaled to the aircraft altitude. To address these issues, we adopt a dynamic spatio-temporal approach in which the satellite pixels with valid AOD retrieval are first co-located within an area of 0.5° x 0.5° centered at aircraft's spatial

location. Although, the selected area of spatial co-location between satellite and aircraft was large, most matchups were found within 0.15° to 0.25° square region for all five cases. In fact, for the SAFARI-2000 case study we found most matchups within 0.10° square region of the aircraft path. The reason for selecting larger box area was to collect as many MODIS retrievals as possible for the comparison. Note that, it is not always necessary that MODIS retrieves ACAOD for all cloudy pixels; some pixels might not have been retrieved due to out-of-domain issue in which the observations fall outside the range of look-up-

table radiances. The airborne AODs are spatially averaged for the consecutive 4 to 5 measurements in order to match with the spatial scale of MODIS retrieval (1 km). The AOD measured by the airborne sensor then is scaled to the area-averaged satellite-retrieved cloud top pressure using information on the vertical distribution of AOD measured during the same flight. This is done by deriving an altitude-dependent AOD polynomial and subsequently using it to estimate columnar AOD at the cloud-top pressure. In this process, it is assumed that the profile of AOD derived from aircraft measurements does not change



during a particular flight and holds its validity over the entire flight path. Finally, the area-averaged AOD measurements from airborne sensor and MODIS are compared. The advantage of scaling the sub-orbital measurements is that both the columnar AOD and the vertical profile of AOD are measured directly, and are therefore considered to be the most reliable in terms of accuracy.

## 3 Results

Figure 1 (top) shows the true-color RGB images captured by Terra/MODIS for different events over three regions where airborne sunphotometer AOD measurements were made above cloud. Super-imposed on these images are the color-coded trajectories of the aircraft where the colors present discrete values of the measured AOD (500 nm). The spatial distribution of ACAOD retrieved from the CR algorithm for these events are shown in the bottom of Figure 1. MODIS retrievals were

10 restricted to pixels having CR<1.05 and COD>3 for better retrieval accuracy (Jethva et al., 2013). Note that the color scale for AOD for both airborne and satellite retrievals is identical. For the SAFARI-2000 event, both the airborne AATS-14 and the MODIS obtain values of ACAOD mostly in the range 0.5-0.6 over the areas north of Walvis Bay, Namibia. MODIS retrievals also show a northward gradient (positive) in AOD, which is likely due to the northern region's proximity to the over-land source of agricultural burning. For the case of Apr 20, 2001 of ACE-ASIA, MODIS retrieves a gradient in AOD along the

15 northern coast of Japan. For the wildfire event observed on Aug 06, 2013 during the SEAC4RS campaign, MODIS retrieves higher values of AOD off the coast of California/Oregon and lower AODs away from the coast.

Figure 2 (left) shows the color-coded scatter plot of ACAOD (500 nm) retrieved from MODIS (y-axis) versus that measured by the airborne AATS/4STAR sensor (x-axis) for the five events of aerosols above clouds. Solid lines represent the expected error

envelope associated with the MODIS retrievals at an actual COD of 10 (Jethva et al., 2013). The statistical summary of the MODIS versus airborne ACAOD comparison is given in Table 2. Among the five events, we find the most extensive matchups accompanied with best comparison for the SAFARI-2000 case of biomass burning aerosols above cloud; this is followed by the dust-above-cloud events observed during ACE-ASIA 2001 and lastly, the wildfire generated carbonaceous aerosols above cloud observed over the Pacific Ocean during SEAC4RS-2013.

The presence of absorbing aerosols above cloud obstructs the light reflected by the cloud top and thus reduces upwelling UV, VIS and SWIR radiation reaching the TOA. Therefore, cloud retrievals, particularly COD, derived from passive sensors such as MODIS are expected to be biased low if absorbing aerosols are not accounted for in the inversion (Wilcox et al., 2009; Jethva et al., 2013). Among several techniques developed to characterize aerosols above cloud, the CR method and the multi-spectral

technique developed by Meyer et al. (2015) retrieve cloud (COD) and aerosol fields (AOD) simultaneously. A comparison plot of COD retrieved by the CR algorithm and that of the MODIS C006 operational product is shown in Figure 2 (right). We find that the CODs retrieved from the CR algorithm are consistently higher by 30-50% than those retrieved from MODIS C006 for the SAFARI-2000 case. For the aerosol events of ACE-ASIA and SEC4RS, CODs retrieved from both algorithms are found to



be comparable in magnitudes. One of the possible reasons for the observed discrepancies between different cases could be the distinct aerosol properties which affects cloud retrieval differently. The negative bias in COD retrieval is directly proportional to the strength of absorption above cloud, which is expressed as the absorption aerosol optical depth (AAOD=AOD*(1-SSA)). Long-term ground-based aerosol inversions made by AERONET over respective regions shows that carbonaceous aerosols

generated from biomass burning over southern Africa are strongly absorbing with SSA (470 nm) of $\sim$ 0.86, whereas aerosols encountered over North-East Asia and western USA during months of the events studied here exhibit relatively weaker absorption capacity (SSA at 470 nm $\sim$ 0.92). For the SAFARI-2000 case, the strong absorption by aerosols above cloud introduces negative bias in the MODIS standard retrieval of COD, as compared with the retrieval by the CR technique that accounts for the aerosol absorption in the cloud retrieval. For the other four cases, it appears that the aerosol absorption is relatively weaker

and likely to be comparable to back-scattering component; thereby making little to no difference in cloud reflected radiation measured at TOA, thus resulting in better agreement between the two independently retrieved cloud retrievals. Second, the departure of the assumed aerosol properties from the actual ones can also lead to error in the COD retrieval. A sensitivity analysis presented in our previous paper Jethva et al. (2013) suggests that for an under-estimated SSA of 0.03 in the assumed SSA yields an error of -4% to -9%, whereas an over-estimation in SSA by the same amount results in an error of about 8% (for

ACAOD of 0.5) to 25% (for ACAOD of 1.0). Note that these are our speculations which demand further analysis supported by the airborne in situ measurements of SSA.

## 4   Discussion and Conclusion

### 4.1   Sources of Uncertainties in ACAOD

Although the satellite retrieval of ACAOD is found to be in good agreement with airborne measurements, some discrepancies

remain. The CR algorithm makes assumptions about the atmosphere and surface in order to perform inversion from satellite observations. Two most important assumptions made in the algorithm are the value of the imaginary part of refractive index, which for a given particle size distribution can be expressed as SSA, and vertical profiles of clouds and aerosols. The theoretical uncertainty analysis adopted for the MODIS wavelengths suggests that, while an uncertainty of $\pm 1$ km in the assumption of aerosol layer height results in errors of 15% or less in ACAOD, the retrievals are found to be more susceptible to the choice of

aerosol model, particularly the SSA (Jethva et al., 2013). For an ACAOD (500 nm) of 1.0, an error of $\pm 0.03$ in the assumption of SSA can lead to an error of about -10/+50% in the ACAOD for an actual COD of 10. In a recent paper by *Meyer et al.*[2015], it is shown that the use of two distinct aerosol models, i.e., one derived from SAFARI-2000 in situ measurements (Haywood et al., 2003) and one adopted from the MODIS dark-target over-land aerosol algorithm MOD04 (Levy et al., 2007), yield different magnitudes of MODIS-based ACAOD over the south-eastern Atlantic Ocean. The two aerosol models differed mainly

in terms of SSA, where the SAFARI-2000 measurements showed SSA of 0.92 and 0.89, and MOD04 aerosol model assumes SSA of 0.88 and 0.80 at 470 and 860 nm wavelengths respectively. Use of these two models resulted in a relative difference of about 30%, 40%, and 60% at an ACAOD of 1.0, 2.0, and 3.0, respectively. The reason for larger errors in ACAOD due to uncertainty in SSA could be following. For an opaque cloud (COD>10) with a fixed value of COD, the changes in TOA





radiances due to aerosols are primarily governed by the aerosol absorption optical depth. Since AAOD is a product of AOD and aerosol co-albedo (1-SSA), a perturbation in SSA from the baseline value with a given fixed value of AAOD will alter the retrieval of ACAOD from its original value. Thus, an erroneous representation of SSA makes algorithm retrieve a different value of AOD in order to explain the observed radiation fields at TOA.

The absorption properties assumed in the aerosol models (Table 1) represent average conditions derived from long-term observations at AERONET sites in the respective regions. These values may vary in space and time, which depends on the type of source, mixing with other pollutants, humidity, and chemical transformation along transport; sometimes the properties also vary systematically (e.g. Eck et al. (2013)). For instance, the SSA retrived by AERONET at *Mongu* on and nearby days around

10 the case study of Sep 13, 2000 was about 0.89 at 470 nm and 0.83-0.86 at 860 nm, which is higher in magnitude compared to the climatological values assumed in the present analysis (Table 1). Similarly, for the ACE-Asia 2001 case studies (Apr, 20, 30, and May 4) the AERONET site at *Noto* reports SSA in the range 0.87 to 0.94 at both wavelengths, whereas the averaged values of SSA assumed in this analysis fall in between this range. During last few year, several techniques have been developed to characterize absorbing aerosols above clouds using NASA's A-train sensors, i.e., OMI (Torres et al., 2012), MODIS (Jethva

et al., 2013; Meyer et al., 2015), POLDER (Waquet et al., 2009), and CALIOP (Chand et al., 2008). Efforts from different groups are underway to apply the respective algorithms on an operational and even near-real time basis. We recognize that it is not possible for an operational aerosol algorithm, be it for cloud-free or above-cloud situations, to characterize aerosol microphysical and optical properties for every pixel on a daily basis. For instance, NASA's operational aerosol algorithms applied to MODIS, OMI, and MISR therefore rely on pre-calculated look-up-tables derived using aerosol-type specific regional

models derived from long-term record of ground-based observations in the respective regions. The main purpose of the 'validation' exercise, such as presented in this paper, is assessing the accuracy of the satellite-based retrievals given the algorithmic assumptions about aerosol models and surface reflectance. For case studies such as presented in the current manuscript, it is expected that using the 'true' values of aerosol microphysical and optical properties for the present case studies will yield more accurate retrievals. However, we emphasize that despite the multi-year approach adopted here for developing aerosol models,

the MODIS retrievals of ACAOD are in good agreement with aircraft measurements within the expected uncertainty limits.

## 4.2 Future validation activities

The present paper has attempted to validate the satellite retrieval of ACAOD using a limited set of airborne sunphotometer measurements. We emphasize here that this work is just the beginning of a continuing exercise of evaluating space-based characterization of aerosols above cloud. Past field campaigns focused on characterizing aerosol properties in cloud-free regions

in order to evaluate and improve satellite-based retrievals, but it left vast cloudy areas unmonitored in terms of the aerosol measurements. Now satellite-based remote sensing techniques using passive sensors are beginning to quantitatively measure aerosol loading above cloud over a large spatial domain; however validation of the these retrievals will remain incomplete without the availability of adequate and accurate airborne measurements.





NASA's ORACLES-ObseRvations of Aerosols above CLouds and their intEractionS (https://espo.nasa.gov/oracles) is an up-coming multi-year field experiment funded by the NASA Earth-Venture Suborbital Program. Beginning in July 2016, the ORACLES experiment intends to make detailed and accurate airborne remote sensing and in situ measurements of the key parameters that govern the cloud-aerosol interaction in the southeastern Atlantic Ocean. Owing to the huge abundance of lofted biomass burning aerosols over the semi-permanent marine boundary layer stratocumulus cloud deck, this region serves as a perfect natural laboratory to assess aerosol-cloud-radiation interactions. Note that this is an area with some of the largest inter-model differences in aerosol forcing assessments on the planet. The experiment will employ a suite of sensors including 4STAR and High Spectral Resolution Lidar (HSRL-2) on NASA's P-3B and ER-2 aircrafts, respectively. Both instruments are capable of making measurements of AOD above cloud and therefore relevant to the assessment of the equivalent satellite retrieval.

In parallel with ORACLES, the Cloud Aerosol Radiation Interactions and Forcing: Year 2016 (CLARIFY-2016) campaign with project partners from the UK Met Office and universities will also take place over the same region with deployment of airborne and surface-based instruments in conjunction with satellite observations of aerosols and clouds. Both of these planned high-profile experiments will deliver a wide range of direct and in situ observations of aerosol above clouds to provide a better process-level understanding of aerosol-cloud-radiation interactions over the SE Atlantic. Among the planned measurements, direct AOD and detailed optical and microphysical measurements of aerosols above cloud will be germane for validating and improving satellite-based retrievals. For instance, the microphysical models, in particular the imaginary part of refractive index and SSA, assumed in the satellite-based inversion pose the largest source of uncertainty in the retrieval. Observations from ORACLES and CLARIFY-2016 will challenge and improve these models for achieving better accuracy in the satellite retrieval.

In addition to the validation activities, inter-comparison of retrievals from A-train sensors should be carried out on various spatial and temporal scales and over distinct hot spot regions of the world, where the overlap of absorbing aerosols and cloud is observed frequently. This is needed to better understand the relative strengths and weaknesses of each sensor and to check the inter-consistency between them. Currently, all ACAOD retrievals are research-only algorithms, but we expect as these algorithms are better understood they could evolve into deliverable operational or semi-operational products on a global scale in the coming years. True validation exercises, such as the opportunities to compare retrievals with a high quality airborne instrument as presented here are essential components in providing the confidence needed towards moving algorithms into operations. A global above cloud aerosol product, in conjunction with standard cloud-free aerosol products derived from satellites, will provide us an unprecedented all-sky aerosol distribution from space. This can substantially enhance our knowledge on how aerosols affect cloud radiative forcing and microphysical properties, and aerosol transport.

*Acknowledgements.* We acknowledge the support of the LAADS team for online availability of MODIS dataset. We also extend our thanks to the PIs of AERONET sites for providing the data that was used to build the aerosol models required for this analysis. The leading author



thanks members of the NASA AATS and 4STAR teams for providing airborne measurements taken during the field campaigns which served as a validation database for the present study.



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



**Table 1.** Aerosol properties assumed in the simulation of TOA reflectance look-up-tables for different events of aerosols above clouds discussed in the text. The assumed microphysical and optical properties were derived from multi-year AERONET measurements at representative sites Mongu (15 °S, 23 °E), Zambia in southern Africa, Noto (37.33 °N, 137.14 °E) in Japan, and HJAndrews (44.24 °N, 122.22 °W). Notations: $R_\mu/R_\sigma$: Mean/standard deviation radius of the Gaussian particle size distribution; $i_{real}/i_{img}$: Real/Imaginary parts of the refractive index; SSA: aerosol single-scattering albedo.

| AERONET Sites | $R_\mu/R_\sigma$ | | $i_{real}$ | | $i_{img}$ | | SSA | |
|---|---|---|---|---|---|---|---|---|
| | Fine | Coarse | 470 nm | 860 nm | 470 nm | 860 nm | 470 nm | 860 nm |
| **Mongu, Zambia** Jul-Aug-Sep 1995-2009 | 0.0898/1.4896 | 0.9444/1.9326 | 1.50 | 1.50 | 0.0262 | 0.0248 | 0.85 | 0.79 |
| **Noto, Japan** Apr 2001-2013 | 0.0886/1.5740 | 0.6036/1.9272 | 1.50 | 1.50 | 0.0092 | 0.0060 | 0.91 | 0.92 |
| **HJAndrews, CA, USA** Aug 1994-2011 | 0.0803/1.5660 | 0.8381/1.9778 | 1.60 | 1.56 | 0.0145 | 0.0165 | 0.92 | 0.86 |

**Table 2.** Statistical summary of the MODIS versus airborne above-cloud AOD comparison.

| Field Campaign | Event Date | Number of Matchups | Root-Mean-Square-Difference | % Matchups Within Predicted Uncertainty (Jethva et al., 2013) |
|---|---|---|---|---|
| SAFARI-2000 | Sept 13, 2000 | 122 | 0.052 | 99.18 |
| ACE-ASIA 2001 | Apr, 20, 30, May 04 | 67 | 0.051 | 83.58 |
| SEAC4RS-20131 | Aug 06, 2013 | 34 | 0.100 | 50.00 |





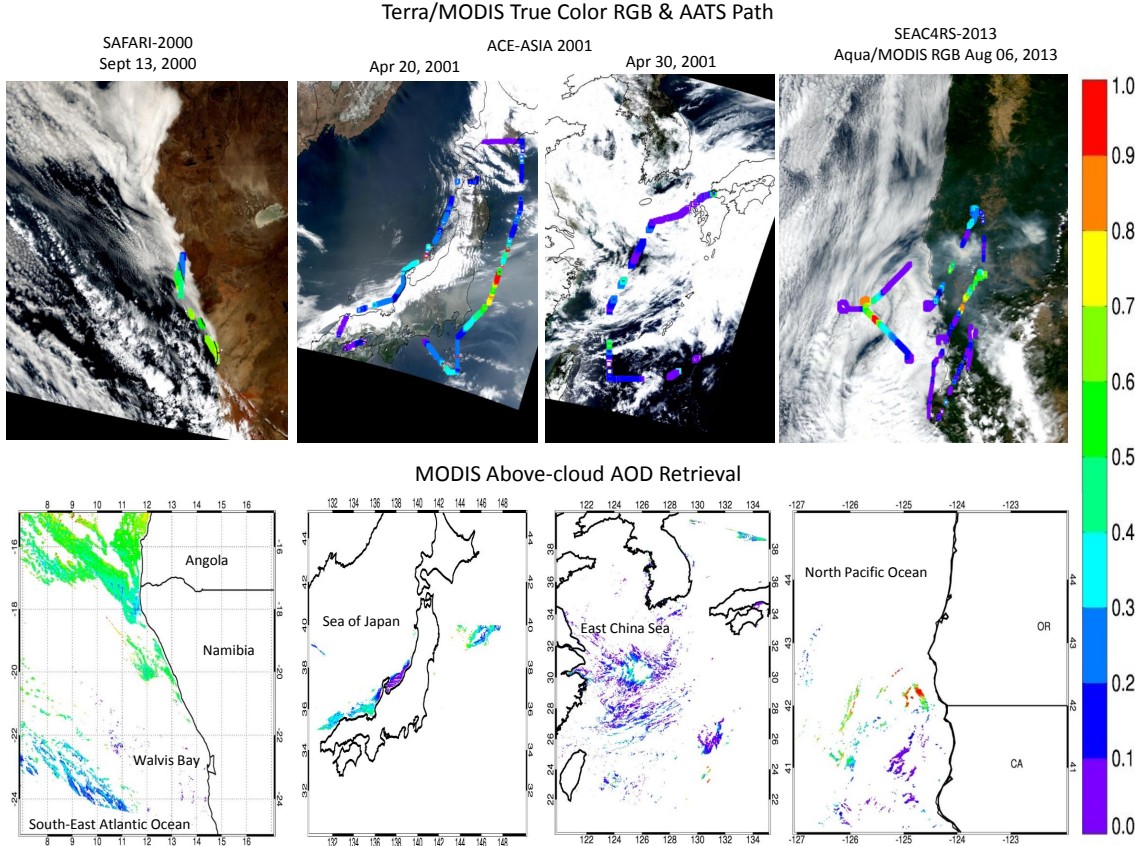

**Figure 1.** (Top) True-color RGB images captured by MODIS superimposed with AOD (500 nm) measured by AATS-6,-14/STAR on Sept 13, 2000, April 20, 30, 2001, and Aug 06, 2013 during SAFARI-2000, ACE-ASIA 2001, and SEAC4RS-2013, respectively. (Bottom) Above-cloud AOD (500 nm) retrieved from MODIS above events.




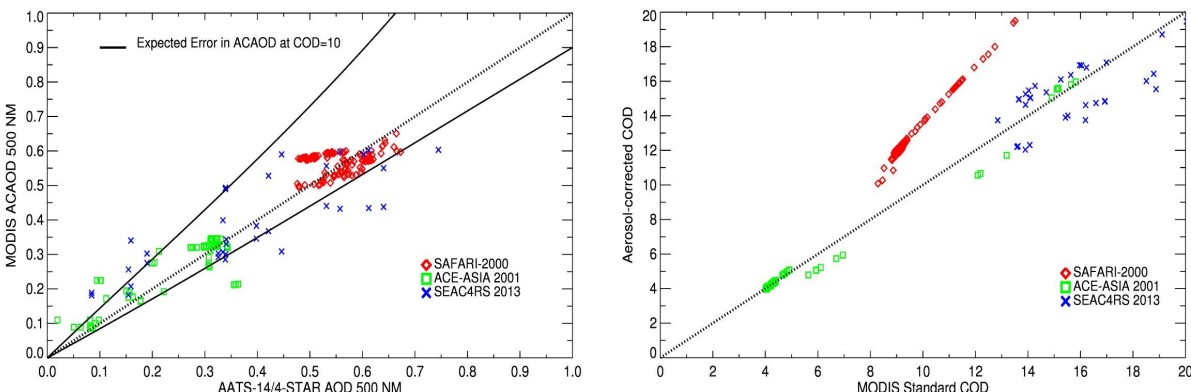

**Figure 2.** (Left) Scatter-plot of above-cloud AOD retrieved from MODIS (y-axis) and that measured by AATS-6,-14/4STAR (x-axis) for the five events discussed in the text. (Right) A comparison of aerosol-corrected COD retrieved from the CR algorithm with those provided by the MODIS standard algorithm (MOD/MYD06).