# Peer review of "Validating MODIS Above-cloud Aerosol Optical Depth Retrieved from 'Color Ratio' Algorithm using Direct Measurements made by NASA's Airborne AATS and 4STAR Sensors"

_Atmospheric Measurement Techniques, 2016_

## Referee Comment (RC1) · Anonymous Referee #3 · 20 Jul 2016

This paper presents an effort to validate a passive satellite imager retrieval of above-cloud absorbing aerosol optical depth using airborne measurements from NASA's AATS and 4STAR instruments. The satellite retrieval is the MODIS color ratio technique developed by the present authors, which uses TOA reflectance at two channels, namely 470 and 860nm, to simultaneously retrieve above-cloud AOD and cloud optical thickness. Comparisons with airborne measurements are shown for five case studies from three field campaigns (SAFARI-2000, ACE-ASIA, and SEAC4RS). The authors show that the MODIS retrievals of above-cloud AOD are in general agreement with the airborne measurements, i.e., a majority of the matchups are within the expected

uncertainty of the MODIS retrievals. In addition, the authors provide a discussion of the challenges that remain for such passive retrievals, focusing specifically on the need to constrain the aerosol radiative models that are at present the largest contributor to retrieval uncertainty.

The paper is well written and provides the sufficient details to understand the analysis. It also represents a significant contribution to the current understanding of aerosol remote sensing, in particular for above-cloud aerosols; as the authors clearly (and rightly) state, this is the first attempt to provide a validation of passive satellite above-cloud aerosol retrievals analogous to the historical efforts to validate clear-sky aerosol retrievals with AERONET. I therefore recommend the paper be accepted for publication in AMT following only minor revisions.

Comments

p. 2, line 19: particular instead of paricularl

p. 2, line 22: The authors here, and elsewhere in the paper, refer to shortwave infrared (SWIR) radiation when referencing what one can infer is the spectral region around 860nm (more obvious references appear later in the paper). Generally speaking, the spectral region from 700 to roughly 1000 or 1100nm is referred to as the near-infrared (NIR), with SWIR referring to wavelengths longer than this but shorter than 3000nm (see, e.g., http://earthobservatory.nasa.gov/Features/FalseColor/page5.php). In the interest of clarity for readers, I suggest the authors verify that their terminology is consistent with general usage, and modify the text accordingly.

p. 5, line 2: "MODIS visible/NIR observations"

p. 5, line 4: I assume the authors use the newest Collection 6 MODIS data? This should be clearly stated.

p. 5, line 34 – p. 6, line 1: Looking at the RGB images in Fig. 1, it appears that the aircraft samples a quite diverse region of the aerosol plumes (e.g., both the middle and

edges), particularly evident in the Apr 20, 2001 ACE –ASIA case and the SEAC4RS case. Is the assumption that the AOD profile is constant throughout the flight therefore valid? It seems to me that the profile could be quite different at plume edge than at plume center. Can the authors comment on this, and perhaps provide the profile statistics for each flight?

p. 7, lines 5 & 7: The authors refer to SSA at 470nm when discussing the absorption effects on the MODIS cloud optical thickness retrievals. However, the MODIS retrievals use either 670nm (over land) or 860nm (over ocean) to retrieve COT. Consider referring to SSA at 860nm instead.

p. 7, lines 9-11: Not only is the aerosol absorption smaller for the radiative models assumed in these cases, but the retrieved AOD is also smaller than what is retrieved in the SAFARI case, which implies a smaller impact on retrieved COT regardless of the aerosol model absorption.

p. 8, line 31: Passive satellite imagers do not "measure" any quantities other than reflected/emitted radiation. All retrievals are therefore derived, or inferred, quantities.

p. 9, lines 19-20: Indeed, constraining the aerosol model is perhaps the most important contribution these campaigns will provide to the passive satellite retrieval science. In my opinion, for these passive above-cloud AOD retrievals, validation efforts such as the one shown here are fundamentally assessments of the aerosol models assumed in the retrievals.

---

## Short Comment (SC1) · 21 Jul 2016

This paper by Jethva et al. compares satellite retrieved aerosol optical depth above clouds with aircraft observations. The authors used Color Ratio method developed by the same group (Torres et al., 2012 and Jethva et al., 2013) for retrieving aerosol optical depth over clouds. Recently, there was a nice effort of inter-comparison of retrieved aerosol optical depth above clouds using multiple-independent methods utilizing passive and active satellites (CALIPSO, OMI, POLDER and MODIS), the comparison among these satellites were very encouraging. So far none of the above cloud

aerosol products from these satellites are cross checked with aircraft or other 'known' observations. Validation of MODIS based CR method using aircraft observations in this study provide additional confidence on the reliability of these new products and very useful for the atmospheric science and climate research. This work demonstrated a potential to deliver new products which are not possible with traditional methods. It also gives a confidence in use of passive satellite observations in presence of clouds. In addition, this paper will also motivate and useful for upcoming campaigns like OR-ACLES and CLARIFY over Africa and its adjoining Atlantic Ocean where the aerosol over the low level clouds are very common in August and September months. Overall the manuscript is well written, introduction is set to nice stage, results are new, conclusive and important and relevant to AMT. I strongly recommend the MS for publication.

---

## Referee Comment (RC2) · A. M. Sayer (Referee) · 23 Aug 2016

I am posting this review under my own name (Andrew Sayer) as I have collaborated with most of the co-authors of this manuscript on this or related topics. I agree with the other reviewers that the manuscript is good and is suitable for publication in AMT after some (mostly minor) revisions, and agree with the specific comments and suggestions that they have made. I'd be happy to review a revised version of the paper if the editor would like.

The one respect in which I disagree with the author (page 3 lines 20-21) and reviewers

is that this isn't the first time an above-cloud AOD product validation has been pub-lished. We did so (using the same SAFARI-2000 and ACE-Asia campaigns) for our own MODIS above-cloud aerosol algorithm in Sayer et al (2016). However as that is quite a new paper, perhaps the reviewers/lead author missed it (it is not cited). So this text should be amended (the abstract makes a similar statement although there it does say it is talking about the colour ratio method). Still, this does not diminish the fact that data set validation is very important and the availability of data to validate this type of algorithm is at present very limited. I strongly support the ongoing collection and analysis of more field data of this type, and congratulate the authors on achieving good performance for the colour ratio algorithm in these case studies.

What is also interesting is that in Sayer et al (2016) we found the same campaign-dependent differences between cloud optical depths from our product vs. the MODIS operational cloud data set (i.e. large positive offset for SAFARI, smaller offset for ACE-ASIA). We also attributed this to differences in aerosol optical properties between the campaigns. Given that our algorithms are distinct (although share similarities), this is perhaps worth mentioning in the revised paper.

I have a general comment about the co-location criteria discussed in section 2.2. The temporal aspect of the matchups was not discussed here. However, in most cases the airborne and satellite data were not collected simultaneously. For example, in the SAFARI-2000 case, the Terra overpass was at 09:25 UTC and the relevant flight seg-ment was from 10:00-13:00 UTC (i.e. between 0.5 and 3.5 hours later). For this specific case we examined the spatiotemporal variability of the AATS and MODIS data, as well as AERONET data affected by the same large-scale smoke event, and concluded that spatiotemporal variability of the aerosol field was sufficiently small that it was probably ok to use this case for validation (it was a very large and fairly homogeneous smoke cloud). Still the idea of doing a pixel-to-pixel spatial match with this time difference is in my view stretching things a little because there will have been some scene changes. For this reason in Sayer et al (2016) we took a box-average for the comparison (i.e. one

point per case study scene) rather than a spatial match along the flight track (i.e. inter-preting the spatial variability on the scales of several pixels and having multiple points in the scatter plot for this scene is probably overanalysing the data in terms of spatial structure, in my view). We reported satellite and AATS mean, median, and standard deviation.

For the 4 May ACE-Asia case, which we also looked at in Sayer et al (2016), the Terra overpass (02:25 UTC) was right in the middle of the flight segment (02:00-03:00 UTC for the cloud we identified) so there was much less temporal mismatch. I am not sure why this case is omitted from the maps in Figure 1? Could it be added, so we can see which cloud(s) were observed and how the retrieval looks?

We investigated but did not use the April 20-30 ACE-Asia cases because the time difference between the flight segments and Terra overpass was too large, so judged that the spatiotemporal variability of the aerosol was too high to use this as a validation case. Looking through my notes, the Terra overpass on April 20 was at 02:10 UTC and the flight was from around 00:00 to 09:00 UTC. Around 02:10 UTC, the plane was around 34.5 N, 140 E, and travelling NE. From Figure 1, there are not many clouds here. The aerosol-laden clouds in the lower panel are in two parts. The area with the clouds on the earlier leg of the flight (around 39 N, 146 E) was overflown around 03:30-04:00 UTC by the aircraft, so about 1.5-2 hours after Terra. The area with the clouds on the later leg of the flight (going from the northern part of the scene down to the SW, going along the northern side of Honshu), was overflown by the aircraft from around 06:00-09:00 UTC, about 4-7 hours after Terra.

For the April 30 flight, the Terra overpass was around 02:50 UTC. The flight was from about 03:10 to 8:20 UTC. The main region with above-cloud aerosols shown in Figure 1 is from about 26 N, 123 E to 32 N, 127 E and was overflown from around 05:45 to 07:30 UTC, around 3 to 4.5 hours after the Terra overpass.

These time mismatches make the idea of using the data points from these two ACE-

Asia flights for validation questionable, in my view, unless there is evidence (as in the SAFARI-2000 case) that the aerosol field was fairly spatiotemporally uniform on these scales. Figure 1 suggests that for these ACE-Asia scenes the AOD field was variable in space and time, based on AATS data, although the sampling is limited. These time mismatches might also explain the increased discrepancies compared to the expected retrieval error seen for some of the ACE-Asia cases in Figure 2 (or it might be coincidental).

We did not analyse the SEAC4RS case from August 6 2013 in Sayer et al (2016) so I'm unsure what the time difference was there, but it should be stated. Did Terra observe the same area? This is the only case study which happened during both Terra and Aqua's lifetimes, so if the timing is right then it would be an interesting opportunity to also compare MODIS Terra vs. Aqua in the presence of the 4STAR validation data.

It is probably fine to keep all these comparisons in the paper, but the time differences for each case and their potential effects should be stated since the measurements are not simultaneous. For 'standard' clear-sky AERONET validation we are fortunate that we are normally able to get near-simultaneous observations (typically within 30 minutes or less), so it is important to point out that we can't always be so lucky here as the reader may not think of this aspect.

Also in section 2.2, the scaling of AOD to the cloud top altitude is an important step. The authors describe their method on pages 5-6 lines 30-1. It would be good to add a figure to illustrate this process for one of the case studies as well, to show the vertical aerosol profile. Is there any estimate of the uncertainty added by this AOD scaling (due to e.g. measurement uncertainty and uncertainty in the satellite-retrieved cloud pressure)?

References:

Sayer A. M., N. C. Hsu, C. Bettenhausen, J. Lee, J. Redemann, B. Schmid, and Y. Shinozuka (2016), Extending "Deep Blue" aerosol retrieval coverage to cases of absorbing aerosols above clouds: Sensitivity analysis and first case studies, J. Geophys. Res. Atmos., 121, 4830–4854, doi:10.1002/2015JD024729.

---

## Author Comment (AC1) · 19 Sep 2016

RC: Referee comments

AR: Author's response

We thank Dr. Sayer for providing his comments and sharing his own thoughts and experience of validating the extended "Deep Blue" above-cloud aerosol retrieval.

RC: About the claim of first validation of the satellite-derived above-cloud AOD

AR: The results presented in this manuscript were actually derived about 2 years ago

and also presented in talks and posters several times in NASA Goddard and elsewhere (2014 AGU Fall Meeting). The manuscript was written last year when no such study existed. A recent paper written by the reviewer (Sayer et al., 2016), who also validates his extended Deep Blue algorithm-based above-cloud AOD retrieval for the same case studies that are shown in our paper. Since the reviewer was fast enough to publish his results earlier than us, we will take back this claim from our manuscript.

RC: About Cloud Optical Depth Comparison

AR: As correctly pointed out by the anonymous referee #3, in addition to the microphysical properties of aerosols, in this case the imaginary part of refractive index and thus single-scattering albedo, the amounts of aerosol loading also plays a determinant role in the resultant bias in the retrieval of cloud optical depth for the above-cloud absorbing aerosols scenes. The AODs for the ACE-ASIA case studies measured/retrieved by AATS/MODIS are relatively smaller in comparison to those of SAFARI-2000. Smaller magnitudes of AOD and higher SSA (less absorption) imply smaller impact on retrieved COT. As stated in the manuscript "the negative bias in COD retrieval is directly proportional to the strength of absorption above cloud, which is expressed as the absorption aerosol optical depth (AAOD=AOD*(1-SSA))." Smaller amounts of aerosol loading with higher SSA yield smaller AAOD and thus result in lesser bias in COT retrieval.

The lower AOD aspect of the bias in cloud retrieval has been discussed in the revised paper.

RC: About the Co-location Criteria and Satellite-Aircraft Time difference

AR: We agree with the Dr. Sayer that the airborne measurements and satellite data were not taken simultaneously. Following is the description of the time differences between MODIS and AATS/4STAR for each flight considered in the validation analysis.

SAFARI-2000 The SAFARI-2000 flight on September 13, 2000 flew between 8 and 14 hours (UTC) over the Walvis Bay, Namibia, whereas Terra/MODIS overpassed the

region at 9:25 (UTC). The co-location procedure yields most matchups between hour 10 and 12 (UTC), leading to a time difference of about 30 minutes to $2\frac{1}{2}$ hours between satellite overpass time and airborne AATS measurements.

ACE-Asia 2001 For the ACE Asia 2001 flights, the time differences between the co-located AATS and Terra/MODIS were about 6 hour, 2-4 hour, and $3\frac{1}{2}$ hours, for the flights operated on Apr 20, 30, and May 04, 2001. For the case of May 04, 2001, we found all matchups (N=38) around Longitude 126°E and Latitude 30°N at hour 6 (UTC), whereas Terra/MODIS flew over the matchup region at 02:25.

SEAC4RS-2013 For the SEAC4RS-2013 flight on Aug 06, 2013, we used MODIS retrievals from both Terra (19:50 UTC) and Aqua (21:30 UTC) platforms for the validation. The time difference between the co-located 4STAR and MODIS observations was from 30 minutes to two hours for Terra, and $\pm$ 2 hours for Aqua. Figure 2 in the original manuscript shows data from Terra and Aqua together. However, the revised Table 2 now shows the statistics for both platforms separately as well as satellite-airborne time difference for each case study.

We agree that the time difference between AATS/4STAR and Terra-Aqua/MODIS overpass for all validation flights considered in the present analysis were larger than in the typical validation exercise of the clear-sky satellite products (30 minutes or even less). However, reviewer should be aware that these are the only major flights of measurements of aerosols above cloud we found after searching the entire past record of AATS and 4STAR. Any changes in the aerosol and cloud fields between the time domain of airborne measurements and satellite retrievals will inevitably introduce mismatch in the comparison.

We have added the time difference attribute of the MODIS vs. AATS/4STAR comparison in section 2.2 "Co-location of Satellite-Airborne Sensors" and also a discussion on the uncertainty in section 4.1 "Sources of Uncertainties in ACAOD" in the revised manuscript.

RC: About Scaling of aircraft-measured AOD to MODIS cloud top pressures

AR: As mentioned in the original manuscript, the absolute error in the aircraft-mounted AATS and 4STAR measurements of AOD for these flights was mostly in the range 0.02 to 0.03 with maximum error reaches up to 0.05 for certain measurements. Assuming the profile of AOD and MODIS-retrieved cloud top pressure are accurate, and therefore scaling itself, the minimum uncertainty expected in the AOD validation analysis would be about 0.03-0.05. A sensitivity study for the SAFARI-2000 case in which the cloud top pressure perturbed by $\pm$ 50 hPa and $\pm100$ hPa from its retrieved values from MODIS resulted in the RMS difference (MODIS minus AATS) of $\pm0.02$ and $\pm0.05$, respectively. These magnitudes of error are comparable to the absolute error in the AATS measurements. The reason for low errors in AOD scaling for this case could be the fact that the AOD measured by AATS between 850 and 900 hPa is almost similar and the MODIS-retrieved cloud top pressure for the matchups points mostly fall in this range of pressure, making not much difference in the scaling even if cloud top pressure is perturbed by $\pm100$ hPa. It is possible that the uncertainty associated with the scaling of AOD would be larger than 0.05 if the sign of error in measurements and cloud retrievals both are on one side, or even can cancel each other if the sign is in opposite direction.

The scaling procedure is now better demonstrated with equations accompanied by a figure for the SAFARI-2000 case in section 2.3 Co-location of Satellite-Airborne Sensors. Also the sensitivity results discussed above have been included in section 4.1 Sources of Uncertainties in ACAOD in the revised manuscript.

**Table 2.** Statistical summary of the MODIS versus airborne above-cloud AOD comparison.

| Field Campaign | Event Date | N | RMSD | % Matchups Within Predicted Uncertainty (Jethva et al., 2013) | ΔT |
|---|---|---|---|---|---|
| SAFARI-2000 | Sept 13, 2000 | 122 | 0.052 | 99.18 | 30 mins to 2 1/2 hours |
| ACE-ASIA 2001 | Apr, 20, 30, May 04 | 67 | 0.051 | 83.58 | 2 to 6 hours |
| SEAC4RS-2013 | Aug 06, 2013 | 34 | 0.100 | 50.00 | T: 30 mins to 2 hours |
| | | T:16 A:18 | T:0.103 A:0.095 | T:35.29% A:61.11% | A: 2 hours |

N: Number of Matchups

ΔT: time difference between satellite overpass and airborne measurements

RMSD: Root-Mean-Square-Difference

T: Terra, A: Aqua

**Fig. 1.** Statistical summary of the MODIS versus airborne above-cloud AOD comparison.

[Figure]

**Fig. 2.** Vertical profile of above-aircraft columnar AOD (left) and ratio of AOD (right) measured by AATS-14 during SAFARI-2000 flight UW1837 flew on September 13, 2000 over the Walvis Bay.

---

## Author Comment (AC2) · 19 Sep 2016

AR: Author's response

AR: We concur with the comments made by Dr. Duli Chand. The validation of the above-cloud AOD is an essential step for understanding the strength and weakness of each independent method developed by different groups, i.e., Chand et al. [2008], Waquet et al. [2008,2013], Torres et al. [2012], Jethva et al. [2013], Meyer et al. [2015], and Sayer et al. [2016]. The validation study, such as presented in our paper, and the ones planned using ORACLES and CLARIFY-2016 field measurements, will provide

much needed confidence and thus credibility for dissemination of such products to broader atmospheric science and climate modeling communities.

---

## Author Comment (AC3) · 19 Sep 2016

RC: Referee comments

AR: Author's response

We thank anonymous referee # 3 for his comments and suggestions that have helped us in improving our manuscript.

RC: p. 2, line 19: particular instead of paricularl

AR: Typo corrected.

[Figure]

RC: p. 2, line 22: About use of SWIR vs NIR terminology

AR: We agree with the reviewer. We used term SWIR loosely here; the 860 nm measurements used in the above-cloud AOD retrieval belongs to near-IR region. SWIR term has been replaced with near-infrared (NIR) throughout the manuscript.

RC: p. 5, line 2: "MODIS visible/NIR observations"

AR: Reviewer is correct; the 'color ratio' technique uses both visible (470 nm) and NIR measurements (860 nm).

Suggestion added in the revised text.

RC: p. 5, line 4: I assume the authors use the newest Collection 6 MODIS data? This should be clearly stated.

AR: Yes, we have used MODIS Collection 006 dataset for all case studies presented in the manuscript. The text in section 2.2 MODIS (under section 2.0 Dataset) is revised as "In the present analysis, we use MODIS Collection 006 products obtained from http://ladsweb.nascom.nasa.gov/data/".

RC: p. 5, line 34 – p. 6, line 1: Looking at the RGB images in Fig. 1, it appears that the aircraft samples a quite diverse region of the aerosol plumes (e.g., both the middle and edges), particularly evident in the Apr 20, 2001 ACE –ASIA case and the SEAC4RS case. Is the assumption that the AOD profile is constant throughout the flight therefore valid? It seems to me that the profile could be quite different at plume edge than at plume center. Can the authors comment on this, and perhaps provide the profile statistics for each flight?

AR: The scaling of AOD from aircraft-level to the cloud top pressure was done using the measured profile of AOD at a particular location during respective flights. While it is possible that vertical structure of aerosols over the cloudy regions away from the measured location may be different, the profile measured by AATS/4STAR is the best educated guess at our disposal to scale the AOD to cloud top pressure. A sensitivity

study for the SAFARI-2000 case in which the cloud top pressure perturbed by $\pm$ 50 hPa and $\pm$100 hPa from its retrieved values from MODIS resulted in the RMS difference (MODIS minus AATS) of $\pm$0.02 and $\pm$0.05, respectively. These magnitudes of error are comparable to the absolute error in the AATS measurements. The reason for low errors in AOD scaling for this case could be the fact that the AOD measured by AATS between 850 and 900 hPa is almost similar and the MODIS-retrieved cloud top pressure for the matchups points mostly fall in this range of pressure, making not much difference in the scaling even if cloud top pressure is perturbed by $\pm$100 hPa. It is possible that the uncertainty associated with the scaling of AOD would be larger than 0.05 if the sign of error in measurements and cloud retrievals both are on one side, or even can cancel each other if the sign is in opposite direction.

The scaling procedure is now better demonstrated with equations accompanied by a figure for the SAFARI-2000 case in section 2.3 Co-location of Satellite-Airborne Sensors. Also the sensitivity results discussed above have been included in section 4.1 Sources of Uncertainties in ACAOD in the revised manuscript.

RC: p. 7, lines 5 & 7: The authors refer to SSA at 470nm when discussing the absorption effects on the MODIS cloud optical thickness retrievals. However, the MODIS retrievals use either 670nm (over land) or 860nm (over ocean) to retrieve COT. Consider referring to SSA at 860nm instead.

AR: The SSA is now referenced at 860 nm in the revised text as follows. Long-term ground-based aerosol inversions made by AERONET over respective regions shows that carbonaceous aerosols generated from biomass burning over southern Africa are strongly absorbing with SSA of âĹij 0.85 and âĹij 0.79 at 470 nm and 860 nm, respectively, whereas aerosols encountered over North-East Asia and western USA during months of the events studied here exhibit relatively weaker absorption capacity (SSA at 860 nm âĹij 0.92 and âĹij 0.86).

RC: p. 7, lines 9-11: Not only is the aerosol absorption smaller for the radiative models

assumed in these cases, but the retrieved AOD is also smaller than what is retrieved in the SAFARI case, which implies a smaller impact on retrieved COT regardless of the aerosol model absorption.

AR: We agree with the reviewer that both smaller magnitudes of AOD and higher SSA (less absorption) imply smaller impact on retrieved COT. As stated in the manuscript "the negative bias in COD retrieval is directly proportional to the strength of absorption above cloud, which is expressed as the absorption aerosol optical depth (AAOD=AOD*(1-SSA))." Smaller amounts of aerosol loading with higher SSA yield smaller AAOD and thus result in lesser bias in COT retrieval.

The lower AOD aspect of the bias in cloud retrieval as suggested by the reviewer has been discussed in the revised paper.

RC: p. 8, line 31: Passive satellite imagers do not "measure" any quantities other than reflected/emitted radiation. All retrievals are therefore derived, or inferred, quantities.

AR: The statement has been revised as "Now satellite-based remote sensing techniques using passive sensors are beginning to quantitatively retrieve aerosol loading above cloud over a large spatial domain"

RC: p. 9, lines 19-20: Indeed, constraining the aerosol model is perhaps the most important contribution these campaigns will provide to the passive satellite retrieval science. In my opinion, for these passive above-cloud AOD retrievals, validation efforts such as the one shown here are fundamentally assessments of the aerosol models assumed in the retrievals.

AR: Since the above-cloud AOD retrievals are most sensitive to the assumption about radiative properties of aerosol model, specifically the imaginary index and thus SSA [Torres, et al., 2012; Jethva et al., 2013; Meyer et al., 2015], the accuracy (or lack thereof) of the satellite-based ACAOD can be interpreted as the assessment of the aerosol models assumed in the inversion. The ORACLES campaign has already been kicked off in Namibia during the last week of August 2016 (https://espo.nasa.gov/oracles). In situ and remote sensing measurements from OR-ACLS of both lofted aerosol layers and cloud beneath will be germane to assess the validity of the algorithmic assumptions and resultant accuracy of the satellite-based ACAOD retrievals.

[Figure]

[Figure]

**Fig. 1.** Vertical profile of above-aircraft columnar AOD (left) and ratio of AOD (right) measured by AATS-14 during SAFARI-2000 flight UW1837 flew on September 13, 2000 over the Walvis Bay.